# Role of MicroRNAs and Long Non-Coding RNAs in Sarcopenia

**DOI:** 10.3390/cells11020187

**Published:** 2022-01-06

**Authors:** Jihui Lee, Hara Kang

**Affiliations:** 1Division of Life Sciences, College of Life Sciences and Bioengineering, Incheon National University, Incheon 22012, Korea; jihui513@naver.com; 2Institute for New Drug Development, Incheon National University, Incheon 22012, Korea

**Keywords:** sarcopenia, non-coding RNA, microRNA, long non-coding RNA, signaling pathway

## Abstract

Sarcopenia is an age-related pathological process characterized by loss of muscle mass and function, which consequently affects the quality of life of the elderly. There is growing evidence that non-coding RNAs, including microRNAs (miRNAs) and long non-coding RNAs (lncRNAs), play a key role in skeletal muscle physiology. Alterations in the expression levels of miRNAs and lncRNAs contribute to muscle atrophy and sarcopenia by regulating various signaling pathways. This review summarizes the recent findings regarding non-coding RNAs associated with sarcopenia and provides an overview of sarcopenia pathogenesis promoted by multiple non-coding RNA-mediated signaling pathways. In addition, we discuss the impact of exercise on the expression patterns of non-coding RNAs involved in sarcopenia. Identifying non-coding RNAs associated with sarcopenia and understanding the molecular mechanisms that regulate skeletal muscle dysfunction during aging will provide new insights to develop potential treatment strategies.

## 1. Introduction

Sarcopenia refers to the progressive and generalized loss of muscle mass, strength, and function, leading to physical disability, fractures, and mortality [1]. It mainly occurs in the elderly due to aging, and it can be a serious health problem [2,3]. From the age of 30 years, muscle mass begins to decline by approximately 1% per year, and the rate of muscle loss accelerates after the age of 70. The prevalence of sarcopenia is 14% among individuals aged 65–70 years, and further increases with age, reaching 53% in those over 80 years of age [4,5]. The loss of muscle mass and strength in the elderly is caused due to progressive physiological changes that disrupt the maintenance of muscle homeostasis. Muscle atrophy, degeneration or senescence of satellite cells, and chronic inflammation have been associated with sarcopenia [6,7,8,9]. Muscle atrophy is defined as the shrinkage of muscle fibers due to the degradation of contractile proteins and organelles by increased proteolysis [10]. Loss of muscle mass is caused by a decrease in the number and size of muscle fibers. Although sarcopenia is primarily a disease affecting the elderly, its progression may be associated with comorbidities that are not limited to the elderly, such as cardiovascular disease, obesity, cancer, and diabetes [11,12,13,14,15]. Many studies on sarcopenia, conducted in diverse models, have reported that catabolic conditions, such as cancer, cachexia, starvation, disuse, menopause, and diabetes, are also accompanied by muscle loss [16,17]. Although sarcopenia is induced by various molecular mechanisms, it is primarily attributed to impaired muscle homeostasis.

Skeletal myogenesis is the process of muscle development wherein satellite cells proliferate, followed by the fusion of differentiated myoblasts into mature myotubes [16,17,18,19]. The process of skeletal myogenesis is regulated by myogenic regulatory factors (MRFs), including myogenic differentiation D (MyoD), myogenic factor 5 (Myf5), myogenin, and myogenic regulatory factor 4 (MRF4) [20]. Satellite cells remain quiescent state and show increased expression of the paired box transcription factor (Pax7), without expressing MyoD [21]. MyoD and Myf5 are involved in the activation of satellite cells from the quiescent state [22,23]. During myogenic differentiation, the expression of Pax protein is suppressed, whereas that of MRF4 and myogenin is upregulated, contributing to myotube maturation [24]. The altered capability of satellite cells to sustain quiescence and their loss of self-renewal and regeneration are major causes of age-related muscle dysfunction and sarcopenia [25]. Aging of satellite cells is characterized by a decrease in stem cell numbers and function and is particularly pronounced in the sarcopenic muscles of humans as well as mice [26]. Since MRFs are essential for muscle development and maintenance of satellite cell functional capacity, alterations in their expression levels contribute to muscle decline and dysregulated muscle physiology in the elderly [27,28].

Age-related sarcopenia is induced by several signaling pathways, including those mediated by transforming growth factor beta (TGF-β), bone morphogenetic protein (BMP), and insulin-like growth factor 1 (IGF-1) [29,30,31,32]. The TGF-β signaling pathway inhibits skeletal muscle growth and induces muscle atrophy [33,34]. Several members of the TGF-β family reportedly play critical roles in regulating muscle physiology [35,36]. For example, TGF-β1, activin A/B, myostatin (growth differentiation factor 8, GDF8), and GDF11 are well-known negative regulators of muscle development and growth [35,37,38]. Upon binding of TGF-β to its receptor (TGF-β receptor), Smad 2/3 is phosphorylated, thereby inducing the translocation of forkhead box O (FOXO) transcription factors. These transcription factors promote the expression of the ubiquitin E3 ligase, atrogin-1, and Muscle RING-Finger 1 (MuRF-1), resulting in protein degradation and muscle wasting [39,40,41]. In addition, phosphorylated Smad 3 interacts with cyclin-dependent kinase (CDK) inhibitors, such as p15, p16, p21, and p27, and induces skeletal muscle senescence [42]. Recent studies have shown that the BMP signaling pathway is a positive regulator of muscle mass [43,44]. BMP 7 binds to BMP receptors and activates the phosphorylation of Smad 1/5/8, which form a transcriptional complex with Smad 4 and inhibit the transcription of ubiquitin E3 ligases, including MuRF-1, atrogin-1, and F-box 30, thereby attenuating muscle atrophy [39,43,44]. Many studies have suggested that alterations in the TGF-β/BMP signaling pathway are involved in muscle differentiation and regeneration of aged muscle [45].

The IGF-1 signaling pathway plays an important role in skeletal muscle growth, differentiation, and regeneration [46,47]. IGF-1 released from myocytes binds to IGF-binding proteins (IGFBPs) and activates the tyrosine kinase receptor. The activated receptor phosphorylates several adaptor proteins, including insulin receptor substrate 1 (IRS-1) and IRS-2, leading to the activation of phosphoinositol 3-kinase (PI3K)-Akt [48]. Subsequently, activated Akt increases muscle mass and strength by inhibiting FOXO and activating mTOR, which induces muscle protein synthesis [49,50]. Decreased IGF-1 expression is reported in experimental models of aging, indicating that sarcopenia is associated with an inhibited IGF-1 signaling pathway [51,52,53].

Non-coding RNAs, including microRNAs (miRNAs) and long non-coding RNAs (lncRNAs), are involved in multiple biological processes and disorders via the regulating several gene networks [54,55]. miRNAs are single-stranded, small non-coding RNA molecules that are 21-25 nucleotides (nts) in length. miRNAs recognize specific target mRNAs through their 3′ untranslated region (3′-UTR) and lead to translational repression or mRNA degradation at the post-transcriptional level [56]. LncRNA transcripts are more than 200 nts in length and can modulate gene expression in various ways, including epigenetic, transcriptional, post-transcriptional, and translational regulation [57]. Recent studies have revealed that non-coding RNAs play critical roles in skeletal muscle development and homeostasis [58,59]. Furthermore, many miRNAs and lncRNAs have been implicated in skeletal muscle dysfunction and muscular diseases, especially sarcopenia [60,61,62]. The differential expression of non-coding RNAs, as observed in models of aging, exacerbates sarcopenia via modulating various signaling pathways [63].

Exercise improves the quality of life by increasing muscle mass, reducing body fat, and enhancing muscle strength and cardiovascular health [64,65]. This implies that exercise exerts preventive and therapeutic effects on muscular dysfunction caused by aging. Several studies have shown that physical activity modulates the expression of non-coding RNAs in the elderly, affecting muscle health and function [66]. Thus, understanding the effects of exercise on non-coding RNA expression during aging can help in developing potential therapeutic strategies to regulate the expression of non-coding RNAs associated with sarcopenia.

In this review, we aimed to summarize the latest information on non-coding RNAs, particularly miRNAs and lncRNAs, associated with sarcopenia. Specifically, we focus on non-coding RNAs that regulate various signaling pathways involved in sarcopenia. In addition, we provide an overview on the expression of non-coding RNAs in response to exercise in context of aging. This review will not only help in understanding the current knowledge regarding sarcopenia pathogenesis mediated by non-coding RNAs, but also guide further research studies and therapeutic development.

## 2. Sarcopenia and Non-Coding RNA Profiling

Advances in transcriptomic technologies and high-throughput analysis allow for the detection of non-coding RNAs, including miRNAs and lncRNAs, that participate in the regulation of gene expression [55]. Several studies have identified miRNAs and lncRNAs implicated in sarcopenia [60,61,62,67,68]. The expression profiles of non-coding RNAs are altered in sarcopenia during skeletal muscle aging [62,69,70,71,72,73,74,75]. Profiling studies on the expression levels of non-coding RNAs during sarcopenia induced by other diseases, such as cancer, have also been conducted [76,77,78,79,80]. Herein, we summarize the information regarding sarcopenia-induced expression of miRNAs and lncRNAs reported so far (Table 1).

### 2.1. miRNAs in Sarcopenia

The majority of studies have identified differentially expressed miRNAs in aging skeletal muscle using genome-wide analyses, such as microarrays or next-generation sequencing (NGS). Hamrick et al., profiled miRNAs in the quadriceps muscles of young (aged 12 months, *n* = 24) and aged (aged 24 months, *n* = 24) mice using microarray [70]. The authors identified that the expression of 57 miRNAs changed with aging in mice, among which the levels of 36 miRNAs were decreased and those of 21 miRNAs increased in aged muscle compared to young muscle. Kim et al., profiled miRNAs in the gastrocnemius muscle of young (aged 6 months, *n* = 6) and aged (aged 24 months, *n* = 6) mice using NGS and found that 34 miRNAs were differentially expressed with age, among which the expression levels of 15 miRNAs were upregulated and those of 19 miRNAs were downregulated. In particular, eight of the downregulated miRNAs were located in a cluster at the imprinted Dlk1-Dio3 locus on chromosome 12 [72].

miRNAs expressed in the tibialis anterior muscle of young (aged 6 months) and aged (aged 24 months) mice were profiled using microarray and NGS [71,74]. Soriano-Arroquia et al., found that the expression of 16 miRNAs was downregulated, while that of 14 miRNAs was upregulated in the muscle during aging using a microarray [74]. Jung et al. used NGS to profile miRNA expression and revealed that the levels of six miRNAs were upregulated while those of 17 miRNAs were downregulated in muscles of aged mice compared to that of young mice [71]. Although the experimental conditions between the two studies were identical, none of the miRNAs were found to be consistently regulated. Furthermore, Jung et al., profiled miRNAs in the tibialis anterior muscle as well as in the serum of young and aged mice, and mice with disuse-induced muscle atrophy (aged 6 months, *n* = 5) that mimics acute atrophy following long-term bed rest [71]. The authors discovered that the expression of only one miRNA (miR-455-3p) was decreased in the muscles of both aged mice and mice with disuse-induced atrophy compared to that in young mice. In serum samples, the levels of two miRNAs were upregulated while those of seven miRNAs were downregulated in aged mice compared to young mice. The expression of only one miRNA (miR-129-5p) was decreased in the serum of aged mice and mice with disused-induced atrophy compared to young mice. In this study, the expression of miR-127-3p and miR-434-3p decreased during aging in muscle tissues and serum [71].

Pardo et al., profiled the miRNAs in the gastrocnemius muscles of young (aged 3 months, *n* = 20) and aged (aged 26 months, *n* = 24) mice using microarray [73]. They discovered that the expression levels of 2 miRNAs were upregulated and those of 13 miRNAs were downregulated in the muscles of aging mice. Microarray-based studies have reported the expression of miRNAs in the quadriceps muscle of young (aged 3 months, *n* = 6) and aged (aged 28 months, *n* = 11) mice [63]. Mikovic et al. found that the levels of 12 miRNAs were downregulated in aged mouse muscle compared to young mouse muscle. Most of them clustered at the Dlk1-Dio3 locus, known as ‘callipyge’ [63]. This region harbors paternally-expressed protein-coding genes, such as delta like non-canonical Notch ligand 1 (Dlk1), retrotransposon gag-like 1 (Rtl1), and deiodinase iodothyronine type III (Dio3), as well as the maternally-expressed non-coding RNA genes, such as maternally-expressed 3 (Meg3 or Gtl2), RNA imprinted and accumulated in nucleus (Rian or Meg8), and antisense Rtl1 (anti-Rtl1). In addition, 54 miRNAs are encoded in this region making it one of the largest miRNA clusters in the genome [81,82]. Several studies have shown that the Dlk1-Dio3 locus is involved in skeletal muscle development [83,84,85]. For example, ovine Dlk1, Gtl2, Rtl1, and Meg8 are abundantly expressed in skeletal muscle and overexpression of Dlk1 and Rtl1 is associated with muscle hypertrophy in sheep [83]. In addition, the expression of Dlk1, Meg3, and Rian as well as a subset of miRNAs from the callipyge locus is increased in mice expressing defective myostatin [84]. Knockdown of Dlk1 reduced the expression of myosin heavy chain II B gene, resulting in a smaller muscle and a lower number of myofibers [85]. Therefore, dysregulation of the callipyge locus may be a potential mechanism underlying age-related muscle decline [63].

Recently, Lee et al. profiled miRNAs in the skeletal muscles of sham and ovariectomized (OVX) mice using microarray [77,85]. Although the OVX model does not exactly recapitulate aging-associated sarcopenia, it exhibits an obesity phenotype along with sarcopenia. In general, sarcopenia and sarcopenic obesity increase after menopause due to decreased muscle strength and increased body fat. In this study, sarcopenia was induced after menopause in young (aged 2 months) mice. The results revealed that seven miRNAs were differentially expressed between the two groups [77]. However, the miRNAs modulated by OVX-induced sarcopenia were not the same as the miRNAs regulated in aging-associated sarcopenia.

In humans, microarray-based miRNA profiles of skeletal muscle from young (aged 31 years, *n* = 19) and aged (aged 73 years, *n* = 17) males have been reported [69]. Drummond et al., discovered that the expression of 40 miRNAs was upregulated and that of 35 miRNAs was downregulated in the skeletal muscle tissues of older adults compared to young adults. miRNA profiles of human muscle tissue from cachectic (*n* = 22) and non-cachectic (*n* = 20) cancer patients using NGS have been reported [78]. Cancer cachexia is characterized by severe depletion of skeletal muscles induced by inflammatory cytokines due to cancer. Narasimhan et al., identified eight miRNAs upregulated during cachexia related muscle wasting [78]. Worp et al., identified 28 differentially expressed miRNAs in patients with cachectic non-small cell lung cancer (NSCLC) compared to healthy controls, among which expression levels of five miRNAs were upregulated and those of 23 miRNAs were downregulated [80]. These two studies revealed that miR-423-3p expression is consistently upregulated in samples of cancer-related cachexia.

Although the degree of aging varied between mouse muscle samples, expressions of miR-127 and miR-434-3p were generally downregulated in most aged mouse muscles. However, so far, none of the studies have reported miRNAs that are commonly upregulated in aged mouse muscles or miRNAs that are consistently modulated in human muscle samples with sarcopenia. Moreover, none of the miRNAs were commonly regulated between human and mouse sarcopenia samples, which might be due to species-related differences or differences in the number of miRNAs between humans and mice. There are approximately 2000 miRNAs encoded by the human genome, whereas approximately 1000 miRNAs are encoded by the mouse genome [86,87]. In addition, approximately 60% of mouse miRNA loci are evolutionarily conserved with those of humans. This may partly account for the abovementioned observations.

### 2.2. LncRNAs in Sarcopenia

While several studies have reported miRNA expression profiles associated with age-related sarcopenia, profiling of lncRNAs involved in sarcopenia is still in early stages. Zhang et al., identified differentially expressed lncRNAs in the gastrocnemius muscles of young (aged 6 months, *n* = 3) and aged (aged 24 months, *n* = 3) mice using an lncRNA array [75]. The authors found 1945 lncRNAs that were aberrantly expressed in aged muscle tissues. Among them, expressions of 894 lncRNAs were downregulated and those of 1051 lncRNAs were upregulated. Sun et al. profiled lncRNAs in the quadriceps muscles of three mouse models of the following catabolic conditions, chronic kidney disease, starvation, and cancer [79]. The expression of eight lncRNAs was elevated, while the expression of 9 lncRNAs was downregulated in atrophied muscles in the three mouse models.

In rats, lncRNA profiles of the femur and quadriceps from OVX and sham models have been reported using NGS [76]. In femur samples, the expression of 17 lncRNAs was altered compared to the sham group. Among them, expression levels of nine lncRNAs were upregulated, whereas those of eight lncRNAs were downregulated. In quadriceps samples, 13 lncRNAs were differentially expressed, of which expressions of eight lncRNAs were upregulated and those of five lncRNAs were downregulated.

Although the experimental conditions that induced sarcopenia varied between mouse muscle samples, the lncRNA 1110038B12Rik was found to be commonly upregulated in catabolic condition-induced atrophic muscles as well as aged mouse muscles. However, in studies conducted to date, none of the lncRNAs are reported to be consistently regulated in sarcopenia between mouse and rat muscles, which might be attributed to species-related differences or a small number of studies.

## 3. Functions of Non-Coding RNAs in Modulating the Signaling Pathways Involved in Sarcopenia

Profiling studies have demonstrated that multiple miRNAs and lncRNAs are involved in aging-induced sarcopenia. Many studies have been conducted to understand the roles of miRNAs and lncRNAs in skeletal muscle biology and the pathogenesis of sarcopenia. In the aging model, the expression levels of non-coding RNAs associated with muscle development are decreased, whereas the expression levels of non-coding RNAs associated with muscle loss are increased. Changes in the expression levels of these miRNAs and lncRNAs affect several signaling pathways and alter target gene expression, contributing to age-related muscle dysfunction. In particular, sarcopenia is exacerbated by activation of the TGF-β signaling pathway, whereas it is alleviated by activation of the IGF-1 signaling pathway, the BMP signaling pathway, and the MRF-related signaling pathway (Figure 1). Here, we summarize the recent reports on the function of miRNAs and lncRNAs that regulate these signaling pathways in sarcopenia pathology (Figure 2).

### 3.1. Non-Coding RNAs That Regulate the TGF-β/BMP Signaling Pathway

miR-431 is one of the downregulated miRNAs in aged myoblasts [88]. Since miR-431 represses Smad 4 expression by directly binding to the 3′-UTR, the decreased levels of miR-431 in aged myoblasts increases Smad 4 expression and consequent activates TGF-β signaling pathway. This results in the inhibition of myogenic differentiation and induction of sarcopenia. Overexpression of miR-431 reduced the inhibition of muscle growth in TGF-β-treated myoblasts [88]. Expression of miR-23a also decreases in a muscle atrophy model [89]. Overexpression of miR-23a repressed Smad 3 expression levels and impaired the phosphorylation of Smad 2/3 in the muscles of diabetic mice, resulting in attenuation of muscle atrophy. miR-23a prevents diabetes-induced muscle cachexia and sarcopenia by inhibiting the TGF-β signaling pathway.

LncChronos is an aging-related lncRNA that is highly expressed in the muscles of aged mice [90]. Inhibition of lncChronos expression activates myofiber hypertrophy both in vitro and in vivo by increasing BMP 7 expression and consequently activating BMP 7 signaling. Aging-induced increase in lncChronos expression exacerbates sarcopenia and muscle atrophy via inactivating BMP 7 signaling.

### 3.2. Non-Coding RNAs That Regulate the IGF-1 Signaling Pathway

Expression of miR-29 is upregulated in aged rodent muscles [91]. It suppresses the expression of IGF-1 by binding to its 3′-UTR, thereby inactivating the IGF-1 signaling pathway. Therefore, aging-induced elevation of miR-29 exacerbates the development of muscle atrophy and sarcopenia [91].

Conversely, in aged satellite cells, miR-143-3p expression is downregulated while that of its target, IGFBP5, is upregulated [92]. Since miR-143-3p inhibits the IGF-1 signaling pathway by targeting IGFBP5, it induces sarcopenia. The age-related alteration in their expression levels may be a compensatory mechanism of protecting the myoblasts from atrophy and improving muscle regeneration. The expression of lncIRS1 is also decreased in atrophied muscles [93]. LncIRS1 inhibits members of the miR-15 family, including miR-15a, miR-15b, and miR-15, which target IRS-1. When expression levels of these miRNAs are downregulated, IRS-1 expression is increased and IGF-1 signaling is activated, which in turn promotes muscle proliferation and differentiation. Decreasing levels of lncIRS1 in the atrophy model induce sarcopenia by impairing the IGF-1 signaling pathway. LncIRS1 could be used as a potential therapeutic agent to treat muscle atrophy.

### 3.3. Non-Coding RNAs That Regulate the MRF-Related Signaling Pathway

In skeletal muscles, lncRNAs act as competing endogenous RNAs for sponging and repressing miRNAs, thereby modulating miRNA targets, and regulating the expression level of MRFs. MRFs are transcription factors that regulate skeletal myogenesis and muscle development [94]. During aging, lower levels or reduced activity of MRF leads to sarcopenia [27,28]. Therefore, controlling the expression of MRFs during aging is important for regulating sarcopenia. LncMAR1 is highly expressed in mouse skeletal muscle and is positively correlated with muscle differentiation and growth both in vivo and in vitro [75]. Increased lncMAR1 expression in mice attenuates muscle atrophy by upregulating the expression levels of myogenic markers, such as MyoD, MyoG, Myf5, and myocyte enhancer factor 2C (MEF2C). LncMAR1 also sponges miR-487b, which targets Wnt5a that is known to stimulate myogenesis and promote muscle differentiation. Downregulation of lncMAR1 expression during aging induces sarcopenia by inactivating the MRF-related signaling pathway and Wnt5a [75]. LncMGPF is a conserved positive regulator of skeletal muscle growth in mice, pigs, and humans [95]. LncMGPF promotes muscle differentiation not only by sponging miR-135-5p, which targets MEF2C, but also by directly interacting with human antigen R (HuR) to stabilize MyoD and MyoG. Although changes in the expression level of lncMGPF remain to be determined in sarcopenia conditions, it was considered to attenuate sarcopenia by inducing myogenic differentiation via activating MRF-related signaling pathways [95].

In contrast, the expression levels of lncDLEU2 were higher in patients with sarcopenia than in those without sarcopenia [96]. LncDLEU2, as a miR-181a sponge, not only upregulates SEPP1 expression, but also decreases the expression levels of MyoD and MyoG, thereby inhibiting muscle differentiation and regeneration. Since lncDLEU2 aggravates sarcopenia by inactivating the MRF-related signaling pathway, it is a potential risk factor for sarcopenia in elderly individuals.

## 4. Exercise and Non-Coding RNAs in Sarcopenia

Since aging-induced sarcopenia affects all skeletal muscles in the body, exercise is vital to improve physical health by increasing muscle mass, enhancing muscle strength and endurance, and boosting physical performance [2,65]. Accordingly, exercise can be considered as a therapeutic strategy for age-related sarcopenia. Indeed, there is growing evidence that exercise contributes to the altered expression of non-coding RNAs in aging individuals [97,98,99,100]. To integrate more information and better understand the role of exercise in non-coding RNA expression during aging-induced sarcopenia, we summarize the studies related to the expression of non-coding RNAs after exercise in a model of aging (Table 2).

### 4.1. miRNAs Modulated by Exercise

Sanctis et al. identified differentially expressed miRNAs and lncRNAs using NGS and showed some of their possible targets and roles through bioinformatics analysis [97]. This study included nine elderly individuals who were trained through resistance training or endurance training and five individuals in a sedentary control group. Resistance training is the anaerobic exercise of muscles against external resistance, such as free weights, weight machines, elastic tubing, or bodyweight, to improve muscle strength, and mass [101,102]. Endurance training, which refers to aerobic exercises such as cycling or running, is performed at low loads over a long period of time and enhances cardiac output and oxygen consumption [103]. Both training methods cause changes in cellular energy metabolism, such as glycolysis and the TCA cycle [104]. Thirty-two miRNAs were reported to be differentially expressed during exercise, among which expression levels of 13 miRNAs were upregulated and those of 19 miRNAs were downregulated [97]. In total, 989 potential target genes, regulated by 32 differentially expressed miRNAs, were involved in cell cycle, cytoskeleton, longevity, and many other signaling pathways. Moreover, 242 lncRNAs were differentially expressed in trained subjects compared to sedentary subjects. Collectively, these studies provide insights into the role of non-coding RNAs involved in the transcriptional networks that modulate the pathways, which counteract aging-associated sarcopenia following exercise [97].

Using microarrays, Zacharewicz et al., demonstrated alteration of miRNA expression in young and aged males after acute resistance exercise [100]. They identified 26 miRNAs that were differentially regulated with age and/or exercise, among which several miRNAs were predicted to affect the Akt-mTOR signaling pathway. Since the Akt-mTOR signaling pathway is involved in the regulation of skeletal muscle mass, these miRNAs may attenuate aging-related sarcopenia.

Interestingly, exercise has also been shown to alter inflammatory cytokine and miRNA expression in sarcopenia [24,105]. In general, plasma levels of pro-inflammatory cytokines, such as tumor necrosis factor α (TNF-α), interleukin 6 (IL-6), and C-reactive protein (CRP), increase with age [106,107]. Aging-induced increased levels of inflammatory cytokines stimulate several molecular pathways involved in skeletal muscle wasting and aggravate muscular strength and hypertrophy [108]. For example, circulating levels of IL-6 and TNF-α, which are linked to loss of muscle strength, are remarkably upregulated in aged individuals with sarcopenia [108]. Rosa et al., demonstrated that improvement of physical condition through rehabilitation exercise in patients with sarcopenia significantly reduced IL-18, IL-37, and CRP levels while increasing miR-355-5p and miR-657 expression, known as post-transcriptional regulators of IL-37 [109]. The authors suggested that IL-37 and its regulatory miRNAs could be used as biomarkers in patients with sarcopenia. These results imply that exercise might alleviate sarcopenia by modulating the expression levels of inflammatory cytokines and miRNAs. In addition, exercise and nutrition are associated with sarcopenia [110]. For example, miR-133b and miR-206 were significantly downregulated in plasma from sarcopenic subjects and correlated with poor nutritional status [110]. These results imply that a nutrient-dependent regulation of miR-133b and miR-206 may play a potential role in age-related muscle decline.

### 4.2. Circulating Exosomal miRNAs Regulated by Exercise

Recently, Nair et al., profiled circulating exosomal miRNAs after acute exercise in aged sedentary males and aged trained males using NGS [99]. Acute exercise changed the expression of circulating exosomal miRNAs in both the groups. In the trained group, 15 exosomal miRNAs were differentially expressed, whereas 13 exosomal miRNAs were differentially expressed in the sedentary group. They discovered that the target genes of the majority of exercise-regulated exosomal miRNAs were involved in IGF-1 signaling, as determined by pathway analysis predictions. While further studies are needed, these results suggest that exercise-induced changes in circulating exosomal miRNAs may regulate the IGF-signaling pathway and prevent age-related muscle loss [99]. Furthermore, Margolis et al. profiled circulating miRNAs using a microarray in young and aged males following resistance exercise [98]. The authors found that the expression of 10 circulating miRNAs was upregulated in young individuals, but downregulated in aged subjects after exercise, and their potential targets were associated with anabolic responses. These results suggest that exercise may induce muscle hypertrophy by regulating circulating miRNAs that stimulate protein synthesis and counteract age-related anabolic resistance [98].

## 5. Discussion

There have been efforts to identify the alterations in the expression levels of non-coding RNAs in patients with sarcopenia. Here, we provide a summary of the studies profiling non-coding RNAs involved in sarcopenia (Table 1). To date, the majority of studies on miRNA profiling have focused on aged mouse models, with reports in humans with small sample sizes. We also noticed that there are no studies on lncRNA using aged human muscle samples. Therefore, more human studies based on advanced transcriptome sequencing technology are warranted to profile miRNAs and lncRNAs involved in aging-related sarcopenia. Further progress in functional studies of non-coding RNAs will provide a basis for understanding skeletal muscle homeostasis and the pathology of sarcopenia. Moreover, establishing standardized diagnostic criteria for sarcopenia models may help classify the progression of sarcopenia and more accurately analyze the molecular mechanisms underlying the pathogenesis of sarcopenia.

In summary, the profiling studies discussed in this review indicate that the expression of miR-127 and miR-434-3p is decreased in aged mouse muscles [63,70,71,72,73,74,77]. miR-127 is reportedly involved in muscle development. Its expression is upregulated during satellite cell differentiation, resulting in enhanced muscle regeneration. The enforced expression of miR-127 in a mouse model of muscular dystrophy ameliorated muscle disease [111]. In addition, miR-127 not only inhibits myoblast proliferation, but also functions as a tumor suppressor in several types of cancer [112,113,114,115]. This implies that miR-127 functions as an activator of cell differentiation and a suppressor of cell proliferation. Interestingly, the expression levels of miR-127 were also downregulated in the brains of aged mice, implying that miR-127 contributes to age-related defects or diseases [116]. Although not many studies have explored the function of miR-434-3p, it has been recently reported that circulating miRNA levels are decreased in patients with sarcopenia accompanied by chronic heart failure [117]. miR-434-3p is downregulated in the skeletal muscle of aging mice and it targets eukaryotic translation initiation factor 5A1 (eIF5A1), resulting in the induction of apoptosis via the mitochondrial apoptotic pathway [73]. This implies that miR-434-3p could be a potential biomarker of skeletal muscle health.

Functional studies have been conducted to examine the role of non-coding RNAs involved in the signaling pathways that regulate sarcopenia [75,88,89,90,91,92,93,95,96]. Since several miRNAs and lncRNAs modulate various signaling pathways associated with sarcopenia pathogenesis, we reviewed recent research studies on non-coding RNA-mediated signaling pathways in sarcopenia (Figure 1 and Figure 2). Non-coding RNAs contribute to muscle loss by regulating the TGF-β/BMP signaling pathway, IGF-1 signaling pathway, and MRF-related signaling pathway. miR-23a, miR-431, lncIRS1, lncMAR1, and lncMGPF promote muscle development and differentiation, whereas miR-29, miR-143-3p, lncChronos, and lncDLEU2 induce muscle atrophy. Although many non-coding RNAs have been identified in profiling studies, their functions and targets have not been fully recognized. Thus, further studies are needed to investigate the direct targets of non-coding RNAs and verify their associated functions in sarcopenia. Furthermore, TGF-β and IGF-1 signaling pathways are interrelated and share some hubs to regulate muscle growth [118]. Several studies have identified a cross-talk between myostatin and Akt/mTOR axis [44]. For example, reduction in Smad 3 phosphorylation due to myostatin blockade activates mTOR signaling and protein synthesis, leading to muscle hypertrophy, which is reversed by rapamycin treatment or mTOR knockdown. It remains elusive whether non-coding RNAs target molecules to regulate these two signaling pathways simultaneously, thereby affecting the phenotypes of sarcopenia.

Recent profiling studies exploring changes in the expression of non-coding RNAs after exercise indicate that exercise has a great therapeutic potential in reducing muscle atrophy and sarcopenia [97,98,99,100]. However, the molecular mechanisms underlying exercise-mediated inhibition of sarcopenia are not fully understood. More validation studies are needed to determine the effect of exercise on the expression of non-coding RNAs that can modulate the expression of target genes. Identifying regulatory non-coding RNAs and their targets and elucidating the molecular mechanisms underlying the effects reversing sarcopenia pathogenesis after exercise will provide new insights for achieving therapeutic benefits against muscle atrophy and muscle dysfunction.

## 6. Conclusions

Sarcopenia is one of the most common age-related conditions and is characterized by progressive loss of muscle mass, strength, and function. Numerous non-coding RNAs are involved in sarcopenia pathogenesis through regulation of the TGF-β/BMP signaling pathway, IGF-1 signaling pathway, and MRF-related signaling pathway. Since non-coding RNAs have great potential as biomarkers or therapeutic targets for sarcopenia, further studies are necessary to elucidate the molecular mechanisms underlying aging-related sarcopenia mediated by non-coding RNAs as well as those of sarcopenia mitigation through exercise. Understanding sarcopenia pathogenesis via non-coding RNA-mediated signaling pathways will not only provide valuable insights to develop therapeutic strategies for sarcopenia, but will also broaden our knowledge of skeletal muscle biology in aging.

## Figures and Tables

**Figure 1 cells-11-00187-f001:**
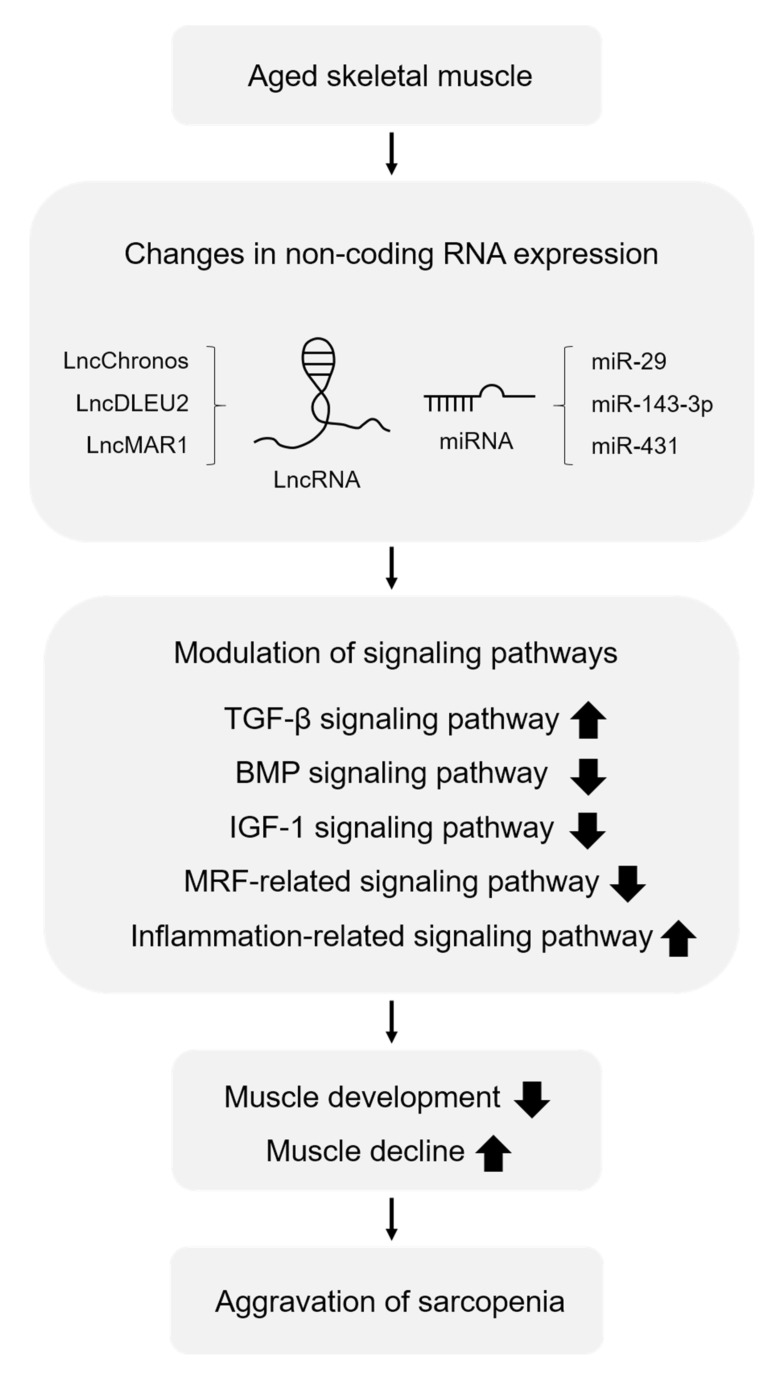
Alteration in non-coding RNA levels regulates multiple signaling pathways in aged skeletal muscle. Sarcopenia is caused by activation of the TGF-β signaling pathway, as well as inactivation of the BMP signaling pathway, IGF-1 signaling pathway, and MRF-related signaling pathway. Non-coding RNAs are involved in these signaling pathways by regulating gene networks and modulating the pathogenesis of sarcopenia.

**Figure 2 cells-11-00187-f002:**
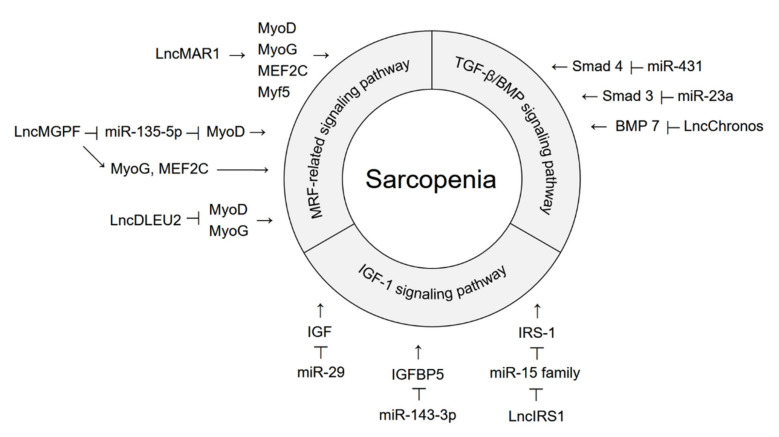
Non-coding RNA-mediated signaling pathways in sarcopenia. The TGF-β/BMP signaling pathway is mediated by miR-431, miR-23a, and lncChronos. miR-29, miR-143-3p, and lncIRS1 are involved in the IGF-1 signaling pathway. LncMAR1, lncMGFP, and lncDLEU2 are involved the MRF-related signaling pathway. Arrow indicates activation of expression; lines with a perpendicular line at the end indicate inhibition of expression.

**Table 1 cells-11-00187-t001:** Studies profiling non-coding RNAs in sarcopenia.

Non-Coding RNA	Species	Sample	Ref.
miRNA	Mouse	Mouse quadriceps muscle young (aged 12 months, *n* = 24) and old (aged 24 months, *n* = 24)	[70]
		Mouse gastrocnemius muscle young (aged 6 months, *n* = 6) and old (aged 24 months, *n* = 6)	[72]
		Mouse tibialis anterior muscle young (aged 6 months, *n* = 3) and old (aged 24 months, *n* = 3)	[74]
		Mouse tibialis anterior muscle and serum young (aged 6 months, *n* = 5) and old (aged 24 months, *n* = 5), and mice with disuse-induced atrophy (aged 6 months, *n* = 5)	[71]
		Mouse gastrocnemius muscle young (aged 3 months, *n* = 20) and old (aged 26 months, *n* = 24)	[73]
		Mouse quadriceps muscle young (aged 3 months, *n* = 6), and old (aged 28 months, *n* = 11)	[63]
		Mouse gastrocnemius muscle OVX and sham group (aged 2 months and left for 15 weeks to induce sarcopenia)	[77]
	Human	Human skeletal muscle young (aged 31 years, *n* = 19) and old (aged 73 years, *n* = 17)	[69]
		Human skeletal muscle cachectic (*n* = 22) and non-cachectic cancer (*n* = 20) patients	[78]
		Human vastus lateralis muscle NSCLC patients with cachexia (*n* = 8), and healthy controls (*n* = 8)	[80]
LncRNA	Mouse	Mouse gastrocnemius muscle young (aged 6 months, *n* = 3) and old (aged 24 months, *n* = 3)	[75]
		Mouse quadriceps muscle chronic kidney disease (CKD), starvation (STV), and cancer	[79]
	Rat	Rat femur and quadriceps muscle OVX and sham group (aged 6 months and left over for 12 weeks to induce sarcopenia, *n* = 12)	[76]

**Table 2 cells-11-00187-t002:** Studies profiling expression of non-coding RNAs in sarcopenia after exercise.

Tool	Non-Coding RNA	Sample and Type of Exercise	Ref.
NGS	miRNA LncRNA	Human vastus lateralis muscle Sedentary, endurance trained, and resistance trained aged males (aged 65–79 years) Endurance and resistance acute exercise bout	[97]
	miRNA	Human plasma Trained (aged 68.2 ± 1.6 years, *n* = 5) and sedentary (aged 70.4 ± 1.4 years, *n* = 5) aged males Endurance acute exercise bout	[99]
Microarray	miRNA	Human skeletal muscle Young (aged 18–60 years, *n* = 10) and aged (aged 60–75 years, *n* = 10) males Resistance acute exercise bout	[100]
		Human serum Young (aged 22 ± 1 years, *n* = 9) and aged (aged 74 ± 2 years) males Resistance acute exercise bout	[98]

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
