# Peer review of "Role of MicroRNAs and Long Non-Coding RNAs in Sarcopenia"

_cells, 2022, doi:10.3390/cells11020187_

Round 1

Reviewer 1 Report

This review summarizes the recent findings of non-coding RNAs associated with sarcopenia and provides an overview of the pathogenesis of sarcopenia through multiple non-coding RNA-mediated signaling pathways. Considering the number of reports published so far, this topic deserves attention. I believe with some major revisions, the current paper would be able to publish and ignite the insights of researchers in the field.

-the manuscript mostly focus on miRNAs and LncRNAs, so the author had better modify the title of the review to highlight the miRNAs and LncRNAs.

-Line 41-42, “Several members of the TGF-β family have been shown to play critical roles in regulating muscle physiology”, please list some more, to make the context more coherent.

-There are some problems of the superscript of the references’ placement in the manuscript, which can lead to readers’ confusion, for example, Line 117-120; 154-155…  Please check and modify the problems carefully.

-In “2.3 miRNAs in sarcopenia” section, Here is a detailed description of the Previous findings of relationship between miRNA and sarcopenia. However, these elements overlap with those in the table.

-“Figure 1”, The picture can further show which kind of miRNAs and lncRNAs has changed with aging or the progress of sarcopenia. The author can further improve the content of the figure.

-In2.1, the gastrocnemius of mice, compared with young mice, most miRNAs in old mice are down-regulated, mainly in the Dlk1-Dio3 locus. I think we can describe in detail the effect of the Dlk1-Dio3 locus on muscle atrophy or sarcopenia.

-In 3.3, LncMAR1 and lncMGPF only describe signal pathways related to MRF. What is the specific process involving MRF pathways? Please add on.

-In the “Exercise and non-coding RNAs in sarcopenia” section,

“The authors suggested that IL-37 and its regulatory miRNAs could be used as biomarkers in patients with sarcopenia” Are there any more references here that show that IL-37 can be used as a biomarker for sarcopenia?

This section I recommend that the authors refer to some previously published relevant reviews (PMID: 34480932, 33293908) to modify and supplement the current framework and content accordingly.

-In the “Discussion” section, “whereas miR-29, miR-143-3p, lncChronos, and lncDLEU2 induce muscle atrophy. ” I want to know whether there is any difference in pathology between muscular atrophy and sarcopenia. Many parts of the article mention muscular atrophy, so is there any difference between muscular atrophy and muscle mass loss?

- Most of the contents in the review is a list of the references, and no relevant personal views are found. Please supplement them.

- The sarcopenia involved in this article is caused by aging, menopause (OVX mouse model), starvation, disuse, cachexia and many other reasons. Is the mechanism consistent and needs to be elaborated? The authors should give more introduction about the sarcopenia.

- The authors cite many review articles in the manuscript, and many of them are still lagging behind. They should cite original articles.

In conclusion, this review requires a certain amount of modification to meet publication requirements.

Author Response

Reviewer 1

This review summarizes the recent findings of non-coding RNAs associated with sarcopenia and provides an overview of the pathogenesis of sarcopenia through multiple non-coding RNA-mediated signaling pathways. Considering the number of reports published so far, this topic deserves attention. I believe with some major revisions, the current paper would be able to publish and ignite the insights of researchers in the field.

  1. the manuscript mostly focus on miRNAs and LncRNAs, so the author had better modify the title of the review to highlight the miRNAs and LncRNAs.

    Response: We agreed with the comment by the reviewer and changed the title to “Role of microRNAs and long non-coding RNAs in sarcopenia”.

  1. Line 41-42, “Several members of the TGF-β family have been shown to play critical roles in regulating muscle physiology”, please list some more, to make the context more coherent.

    Response: As the reviewer suggested, we mentioned that members of the TGF-β family, such as TGF-β1, activin A/B, myostatin (growth differentiation factor 8, GDF8), and GDF11, are well-known negative regulators of muscle development and growth in the introduction section (line 108-110).

  1. There are some problems of the superscript of the references’ placement in the manuscript, which can lead to readers’ confusion, for example, Line 117-120; 154-155…  Please check and modify the problems carefully.

    Response: We agreed with the comment by the reviewer. We checked and corrected the references in order not to confuse the reader.

  1. In “2.3 miRNAs in sarcopenia” section, Here is a detailed description of the Previous findings of relationship between miRNA and sarcopenia. However, these elements overlap with those in the table.

    Response: We agreed with the comment by the reviewer. We wanted to present a tabular summary of all profiling studies of non-coding RNA in sarcopenia covered in section 2 (Sarcopenia and non-coding RNA profiling) for easier understanding.

  1. “Figure 1”, The picture can further show which kind of miRNAs and lncRNAs has changed with aging or the progress of sarcopenia. The author can further improve the content of the figure.

    Response: As the reviewer suggested, we changed Figure 1 in the revised manuscript.

  1. In2.1, the gastrocnemius of mice, compared with young mice, most miRNAs in old mice are down-regulated, mainly in the Dlk1-Dio3 locus. I think we can describe in detail the effect of the Dlk1-Dio3 locus on muscle atrophy or sarcopenia.

    Response: We agreed with the comment by the reviewer and described studies on the effect of the Dlk1-Dio3 locus on skeletal muscle development in the 2.1 section (line 658-672).

  1. In 3.3, LncMAR1 and lncMGPF only describe signal pathways related to MRF. What is the specific process involving MRF pathways? Please add on.

    Response: As the reviewer suggested, we described skeletal myogenesis regulated by MRFs in the Introduction section (line 45-102) and mentioned the importance of the control of the MRF expression in sarcopenia in the 3.3 section (line 1089-1092).

  1. In the “Exercise and non-coding RNAs in sarcopenia” section,

“The authors suggested that IL-37 and its regulatory miRNAs could be used as biomarkers in patients with sarcopenia” Are there any more references here that show that IL-37 can be used as a biomarker for sarcopenia?

Response: We tried to look for more references, but couldn’t find any more.

  1. This section I recommend that the authors refer to some previously published relevant reviews (PMID: 34480932, 33293908) to modify and supplement the current framework and content accordingly.

    Response: As the reviewer suggested, we modified the framework of section 4.

  1. In the “Discussion” section, “whereas miR-29, miR-143-3p, lncChronos, and lncDLEU2 induce muscle atrophy. ” I want to know whether there is any difference in pathology between muscular atrophy and sarcopenia. Many parts of the article mention muscular atrophy, so is there any difference between muscular atrophy and muscle mass loss?

    Response: Muscle atrophy, degeneration or senescence of satellite cells, and chronic inflammation have been associated with sarcopenia. Muscle atrophy is defined as the shrinkage of muscle fibers due to the degradation of contractile proteins and organelles by increased proteolysis. Loss of muscle mass is caused by a decrease in the number and size of muscle fibers. We added about this in the introduction section (line 30-36).

  1. Most of the contents in the review is a list of the references, and no relevant personal views are found. Please supplement them.

    Response: We analyzed the results of profiling studies on aged mouse models reviewed in this review. We found that miR-127 and miR-434-3p generally decreased in aged mouse muscles and this was discussed in the Discussion section of the original manuscript (line 1568-1583). But as the reviewer suggested, we tried to add a little more relevant personal view in the Discussion of the revised manuscript (line 1558-1567).

  1. The sarcopenia involved in this article is caused by aging, menopause (OVX mouse model), starvation, disuse, cachexia and many other reasons. Is the mechanism consistent and needs to be elaborated? The authors should give more introduction about the sarcopenia.

    Response: Although sarcopenia is primarily a disease affecting the elderly, its progression may be associated with comorbidities that are not limited to the elderly, such as cardiovascular disease, obesity, cancer, and diabetes. Many studies on sarcopenia, conducted in diverse models, have reported that catabolic conditions, such as cancer, cachexia, starvation, disuse, menopause, and diabetes, are also accompanied by muscle loss. Although sarcopenia is induced by various molecular mechanisms, it is primarily attributed to impaired muscle homeostasis. As the reviewer suggested, we added about this in the introduction section (line 36-42).

  1. The authors cite many review articles in the manuscript, and many of them are still lagging behind. They should cite original articles.

    Response: As the reviewer suggested, we checked the references and cited 29 more original articles.

Reviewer 2 Report

The authors provided a summary of the profiling studies on non-coding RNAs involved in sarcopenia. 

The article can be accepted after major revisions.

My suggestions are the following:

  • the majority of studies on miRNA profiling have been focused on aged mouse models, with reports in humans realized in studies with a small sample size. The authors should comment on this point. they say that there are specie-specific differences among miRNA in human and mouse, or mouse and rats, so please they should comment this point of conservation along evolution. Do they expect conversation or not? And why?
  • Signaling pathways like TGFbeta and IGF1 are interrelated among them and share some hubs. The authors should comment on this.
  • From line 38 to 64 the paragraph on molecular mechanism can be resumed in a box and a figure, representing the pthways and how they interact, as suggested at point 2.
  • In table 1, it should be useful to have a list of relevant miRNA to compare the different studies and find consistency
  • In figure 1, in the scheme is missing the inflammation. it is a predisposing factor to muscle wasting and finally ending in sarcopenia. literature data report that inflammation and oxidative stress induce expression of myostatin, a member of the TGF-β superfamily that inhibits skeletal muscle growth muscle mass by downregulating the expression of miR-133b and miR-206 in skeletal muscle. this part is discussed afterwards but I will put it here in the scheme
  • please define the differences between endurance exercise and resistance exercise in terms of metabolism

Author Response

Reviewer 2

The authors provided a summary of the profiling studies on non-coding RNAs involved in sarcopenia. 

The article can be accepted after major revisions.

My suggestions are the following:

  1. the majority of studies on miRNA profiling have been focused on aged mouse models, with reports in humans realized in studies with a small sample size. The authors should comment on this point. they say that there are specie-specific differences among miRNA in human and mouse, or mouse and rats, so please they should comment this point of conservation along evolution. Do they expect conversation or not? And why?

    Response: We agreed with the comment by the reviewer and mentioned that the majority of studies on miRNA profiling have focused on aged mouse models, with reports in humans with small sample sizes in the Discussion section (line 1558-1560). We also added possible reasons why miRNAs that are commonly regulated in humans and mice, or mice and rats have not been elucidated in section 2.1 (line 699-933). It could be due to species-related differences or differences in the number of miRNAs between humans and mice. There are approximately 2,000 miRNAs encoded by the human genome, whereas approximately 1,000 miRNAs are encoded by the mouse genome. In addition, approximately 60% of mouse miRNA loci are evolutionarily conserved with those of humans. This may partly account for the abovementioned observations.

  1. Signaling pathways like TGFbeta and IGF1 are interrelated among them and share some hubs. The authors should comment on this.

    Response: As the reviewer suggested, we mentioned the interrelation of TGF-β and IGF-1 signaling pathways in the Discussion section (line 1864-1870).

  1. From line 38 to 64 the paragraph on molecular mechanism can be resumed in a box and a figure, representing the pthways and how they interact, as suggested at point 2.

    Response: As in the answer to comment #2, we mentioned the interrelation of TGF-β and IGF-1 signaling pathways in the Discussion section.

  1. In table 1, it should be useful to have a list of relevant miRNA to compare the different studies and find consistency

    Response: We agreed with the comment by the reviewer, but the number of miRNAs that changed in sarcopenia in each study was too large to be presented in a table. If there is a miRNA whose function was validated in each study, we mentioned in section 2.1 and presented in Figure 1 of the revised manuscript. In addition, we analyzed the results of profiling studies on aged mouse models reviewed in this review. We found that miR-127 and miR-434-3p generally decreased in aged mouse muscles and this was discussed in the Discussion section (line 1568-1583).

  1. In figure 1, in the scheme is missing the inflammation. it is a predisposing factor to muscle wasting and finally ending in sarcopenia. literature data report that inflammation and oxidative stress induce expression of myostatin, a member of the TGF-β superfamily that inhibits skeletal muscle growth muscle mass by downregulating the expression of miR-133b and miR-206 in skeletal muscle. this part is discussed afterwards but I will put it here in the scheme

    Response: As the reviewer suggested, we added the inflammation-related signaling pathway into the scheme of Figure 1.

  1. please define the differences between endurance exercise and resistance exercise in terms of metabolism

    Response: As the reviewer suggested, we mentioned the differences between endurance exercise and resistance exercise in section 4 (line 1329-1335).

Round 2

Reviewer 1 Report

Accept in present form.

Reviewer 2 Report

The authors addressed to my concerns, with the ecception of one, due to the large number of miRNA to be put in the Table. They can try to put in a supplementary table. Anyway,  I can accept it.